# When cheating turns into a stabilizing mechanism of plant–pollinator communities

**François Duchenne**[1]*, **Stéphane Aubert**[1], **Elisa Barreto**[1], **Emanuel Brenes**[2], **María A. Maglianesi**[2], **Tatiana Santander**[3], **Esteban A. Guevara**[1], **Catherine H. Graham**[1]

**1** Swiss Federal Institute for Forest, Snow and Landscape Research (WSL), Birmensdorf, Switzerland, **2** Escuela de Ciencias Exactas y Naturales, Universidad Estatal a Distancia (UNED), San Pedro de Montes de Oca, San José, Costa Rica, **3** Aves y Conservación (BirdLife Ecuador), Quito, Ecuador

* francois.duchenne@wsl.ch

**Data Availability Statement:** Data and R codes are available here: https://doi.org/10.5281/zenodo.10102438 R codes are also available here: https://

## Abstract

Mutualistic interactions, such as plant–mycorrhizal or plant–pollinator interactions, are widespread in ecological communities and frequently exploited by cheaters, species that profit from interactions without providing benefits in return. Cheating usually negatively affects the fitness of the individuals that are cheated on, but the effects of cheating at the community level remains poorly understood. Here, we describe 2 different kinds of cheating in mutualistic networks and use a generalized Lotka–Volterra model to show that they have very different consequences for the persistence of the community. Conservative cheating, where a species cheats on its mutualistic partners to escape the cost of mutualistic interactions, negatively affects community persistence. In contrast, innovative cheating occurs with species with whom legitimate interactions are not possible, because of a physiological or morphological barrier. Innovative cheating can enhance community persistence under some conditions: when cheaters have few mutualistic partners, cheat at low or intermediate frequency and the cost associated with mutualism is not too high. Under these conditions, the negative effects of cheating on partner persistence are overcompensated at the community level by the positive feedback loops that arise in diverse mutualistic communities. Using an empirical dataset of plant–bird interactions (hummingbirds and flowerpiercers), we found that observed cheating patterns are highly consistent with theoretical cheating patterns found to increase community persistence. This result suggests that the cheating patterns observed in nature could contribute to promote species coexistence in mutualistic communities, instead of necessarily destabilizing them.

## Introduction

Mutualism, an interaction that benefits all species involved, is widespread in the web of life [1,2]. In natural systems involving mutualism, it is common to find cheaters, species reaping benefits from interactions without providing benefits to their partners [3,4]. Cheating has been described in mycorrhizal [5], plant–pollinator [6], cleaner fish-client [3], and seed dispersal [7] interactions. Evolutionary theory has shown that a stable coexistence of mutualists

github.com/f-duchenne/Cheating_in_mutualistic_networks.

**Funding:** The material for the project and the time spent collecting and analysing data were funded by two different grants: one from the European Research Council (ERC, https://erc.europa.eu/homepage) under the European Union's Horizon 2020 research and innovation program (grant agreement N°787638, granted to C.H.G.) and one from the Swiss National Science Foundation (N° 173342, granted to C.H.G., https://www.snf.ch/en). The funders had no role in study design, data collection and analysis, decision to publish, or preparation of the manuscript.

**Competing interests:** The authors have declared that no competing interests exist.

and cheaters is possible [8–10], explaining their persistence in nature. Using pollination as a study model, it has been shown that illegitimate interactions between pollinators and plants (cheating, i.e., nectar robbing) affect the structure of the interaction networks [6,11]; however, the consequences of cheating for the ecological stability of these communities have been largely overlooked, but see [12].

From an evolutionary perspective, widely based on game theory, cheating is only viewed as a way to escape the cost associated with mutualism [8,13–15], for example, energy expenditure due to nectar production in plants or foraging in animals. In this case, while the species could interact legitimately (i.e., mutualistically), they cheat because to them the cost of cheating is lower than the cost of legitimate interactions. However, species can also cheat to profit from partners that are otherwise unavailable, when a legitimate interaction is not possible because of a physiological or morphological barrier [16,17]. In the first case, cheating is directly detrimental for the mutualism because it replaces mutualistic interactions with antagonistic or commensalistic ones (Fig 1A). We term this type of cheating, where the interaction niche of the species remains constant (Fig 1B), conservative cheating. In the second case, cheaters use a partner that is otherwise unavailable, thus resulting in new interactions that would not exist in a purely mutualistic framework (Fig 1A). We term this second type of cheating, where the interaction niche of the cheater expands (Fig 1B), innovative cheating.

We reasoned that 2 main factors should mediate how cheating impacts the number of species being able to coexist (i.e., network or community persistence). First, increasing cheating frequency reduces mutualistic benefits, leading to reduced persistence of the cheated species, which in turn decreases the overall network persistence. Second, low or intermediate cheating frequencies can help cheaters to persist, which can feedback into more mutualistic partners for the other guild and thereby increasing network persistence.

We expect that the balance between these 2 opposite effects on network persistence depends on whether cheating is conservative or innovative, as the type of cheating will reorganize the structure of interactions in different ways. When conservative cheating increases, we expect a decrease in network persistence. This is because conservative cheating turns some mutualistic interactions, which tend to increase persistence [18], into antagonistic or commensalistic interactions, without species gaining additional partners. For innovative cheating, our expected outcome on network persistence is uncertain. Innovative cheating also introduces antagonistic or commensalistic interactions in mutualistic networks, but cheaters can rely on more partners potentially increasing their persistence, and thus the overall network persistence.

Consequently, when cheating is innovative we expect different effects according to the role of the cheaters. If cheaters are generalist species (i.e., have many mutualistic partners), we expect that their persistence will be only slightly enhanced by innovative cheating, as they already have many partners. In this case, innovative cheating is more likely to decrease benefits for mutualistic partners with few positive counterparts, leading to a decrease in network persistence. If cheaters are specialists (i.e., with few partners), we expect that innovative cheating increases the overall network persistence because it will increase the persistence of specialist species. Specialists are often the most vulnerable species to extinction because of competitive exclusion [19,20], thus increasing the persistence of specialists could increase the overall network persistence through positive feedback loops arising in mutualistic networks. However, if innovative cheating is too important, the resulting niche expansion of specialist cheaters (Fig 1B) could turn these specialists into generalist species and promote competitive exclusion of other species from the same guild. Thus, we expect that a low or intermediate frequency innovative cheating would increase network persistence by protecting specialist species, but that a high frequency innovative cheating would decrease network persistence by promoting

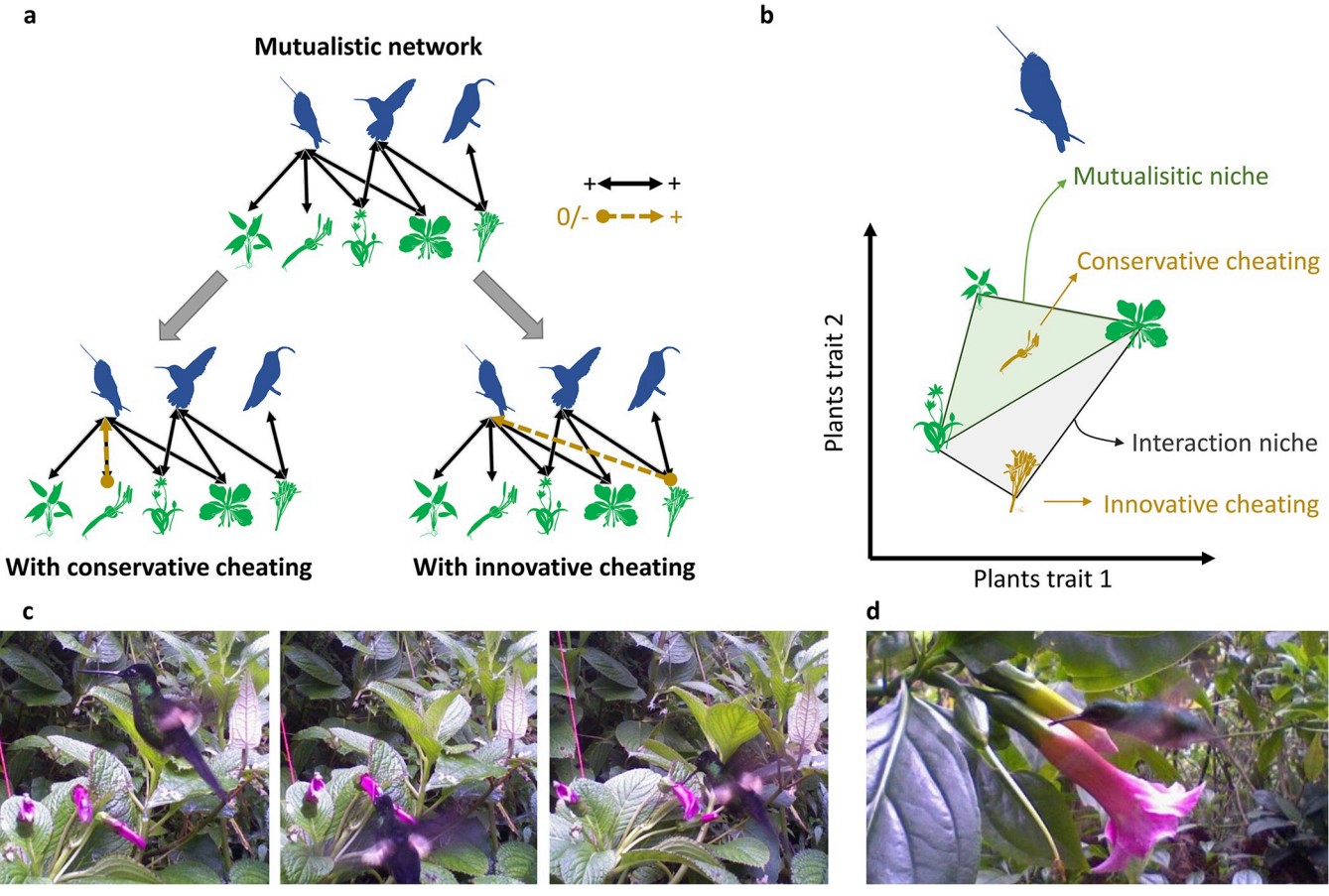

**Fig 1. Cheating can be conservative or innovative.** (a) Above, mutualistic network built using only +/+ (legitimate) interactions and below, mutualistic networks including 1 illegitimate interaction. Cheating is conservative if it happens between partners that were already linked by a mutualistic interaction, or innovative if it happens between partners that were not linked by a mutualistic interaction. (b) Cheating characterized in a niche concept. Symbols represent the position of the legitimate (green) and illegitimate (yellow) partners of a given species of pollinator in their trait space. Conservative cheating occurs within the mutualistic niche, which is the convex hull determined by trait combination of legitimate partners. Innovative cheating occurs with plants that are outside of the mutualistic niche and lead to an expansion of the interaction niche, defined as the convex hull determined by trait combination of legitimate and illegitimate partners. (c) An example of conservative cheating, because the same species of hummingbird (*Coeligena lutetiae*) visits legitimately (middle) and illegitimately (right) flowers of *Centropogon pichinchensis*. (d) An example of innovative cheating, where *Eupherusa eximia* visits illegitimately a flower of *Symbolanthus pulcherrimus*, which has a corolla much longer than the bill of the hummingbird, preventing legitimate interaction.

competitive exclusion. Since these indirect facilitation and competition effects arise mainly in large networks [21,22], the effects of cheating would be fully assessable only if considering a realistic diversity, but not when considering a module of few species.

In empirical systems, cheating is often evidenced indirectly, for example, using the presence of holes in flowers [23,24]. Such data do not allow evaluation of cheating at the interaction level, as the identity of the cheaters is unknown. However, indirect evidence suggests that innovative cheating is common is pollination, because pollinators often rob nectar in flowers with long corolla, in which nectar is difficult to access [16,24,25]. Thus, in addition to considering different kinds of cheating in theoretical models, we also need to collect empirical data about the relative distribution of legitimate and illegitimate interactions in mutualistic networks, to understand the consequences of cheating at the community level.

To test the hypotheses developed above, we used a theoretical ecological model, without evolutionary dynamics, to assess the consequences of conservative and innovative cheating for the ecological stability of bipartite mutualistic networks, measured as network persistence. We

used a generalized Lotka–Volterra model with 2 guilds with a moderate diversity, 20 plant and 20 pollinator species, embedded in an obligatory mutualistic relationship. We incorporated 3 parameters to model cheating: the proportion of pollinator species cheating ($\bar{\Delta}$), the cheating frequency of these cheaters ($\Omega$), the proportion of innovative cheating versus conservative cheating ($\Psi$); and a fourth parameter to model the cost associated with mutualism ($\Lambda$), that is likely to mediate the effect of cheating. For simplicity, we assumed that only pollinators cheat, so only they were able to interact illegitimately with plants, i.e., piercing the flower corolla and robbing the nectar without pollinating. In this model, cheaters escape the cost associated with mutualism while robbed plants do not benefit from these interactions and still pay the cost associated with mutualism (i.e., nectar production). We also assumed no additional cost for plants due to cheating than the one associated with mutualism, as cheaters often slightly damage flowers. To test how the degree of generalism of the cheaters could affect community persistence, we performed simulations for 2 scenarios: when the cheaters were the most generalist species and when they were the most specialist species in the community (*cf*. Methods).

Using these results, we identified conditions in which cheating increases ecological persistence of communities. In a second step, to see if these conditions are met or not in nature, we used an empirical dataset including approximately 34,000 interactions between flowering plants and hummingbirds collected in Costa Rica and Ecuador. Each interaction was classified as legitimate or illegitimate, allowing us to explore cheating patterns in empirical mutualistic networks. Finally, to close the loop between theory and data, we used these empirical data to parameterize our model and assess how observed patterns of cheating affect network persistence.

## Results

Our simulations showed that the output of cheating on network persistence, measured as the percentage of persisting species at equilibrium, could be negative or slightly positive (Fig 2A and Fig A in S1 Results) and strongly depended on its conservative versus innovative character.

As expected, when cheating was conservative it almost always negatively affected network persistence, because cheating decreases benefits of plant–animal interactions for plants, thus decreasing their persistence. In an obligatory mutualism, interacting guilds depend on each other; therefore, a decrease in plant persistence is likely to negatively affect overall network persistence. Regardless of its conservative/innovative character, cheating had a more negative (or less positive) effect on persistence when there was a cost associated with mutualism ($\Lambda > 0$, Fig 2A), i.e., when illegitimate interactions are antagonistic. Similarly, on average, cheating had a less negative (or more positive) effect on network persistence when cheaters were specialist species than when they were generalists (Fig 2A). This difference is because cheating makes specialists better able to persist and thus can buffer the negative effects of cheating on their partners through positive feedback loops arising in mutualistic networks (Fig B in S1 Results). In contrast, cheating had few positive effects on persistence of generalist species (Fig B in S1 Results), so when they cheat, cheating is likely to negatively affect network persistence.

The window of parameters in which cheating increases network persistence, relative to cases with no cheating, mostly correspond to the cases when: cheaters were specialists, cheating was innovative with intermediate frequencies, and the cost associated with mutualism was low (Fig 2). This parameter space can be represented in 3 dimensions (Fig 2C), using the 3 parameters controlling for cheating distribution, the proportion species cheating ($\bar{\Delta}$), the cheating frequency of these cheaters ($\Omega$), and the proportion of innovative cheating ($\Psi$). The

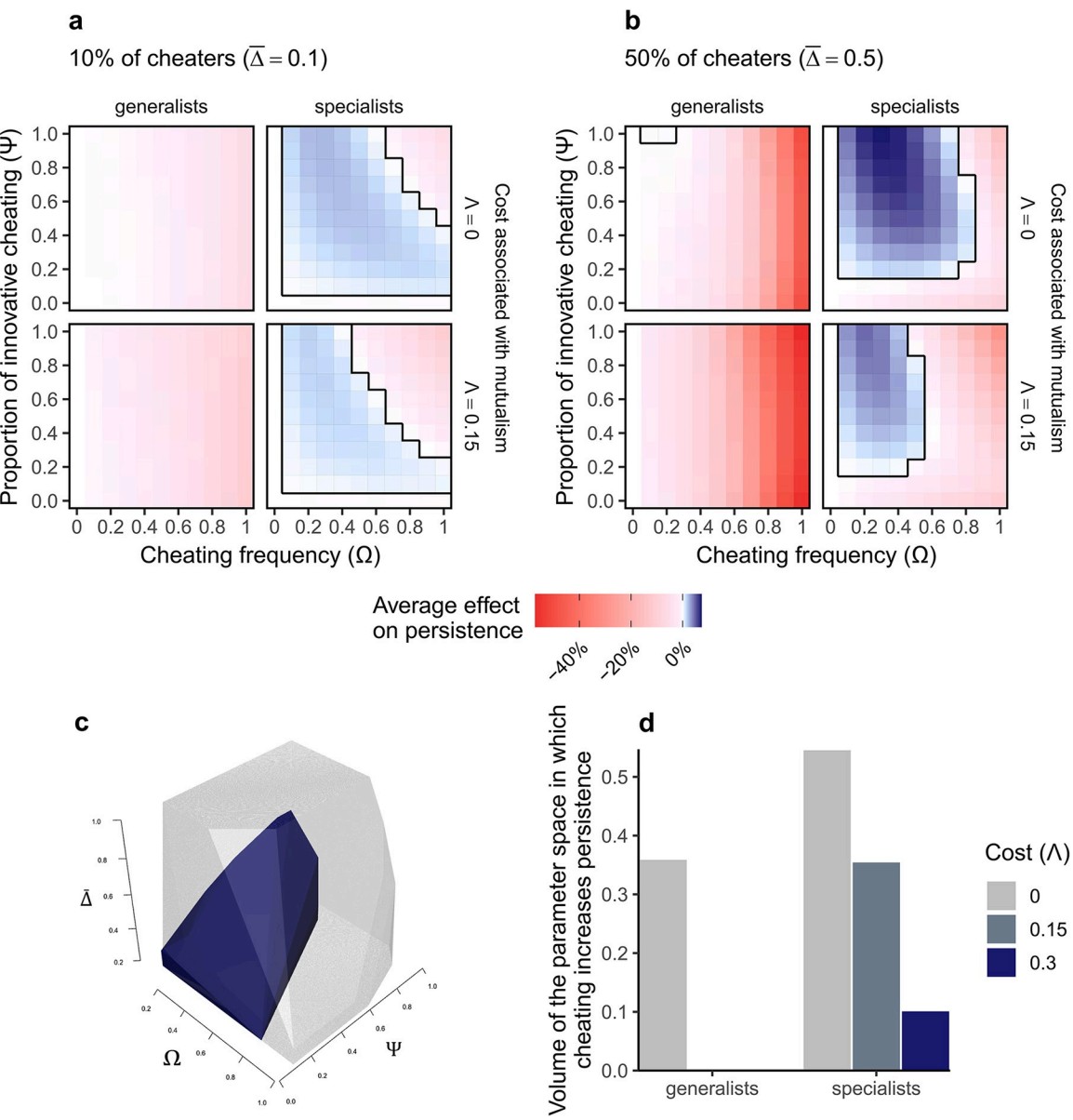

**Fig 2. Innovative cheating can increase network persistence.** Network persistence (percentage of persisting species at equilibrium) as a function of cheating frequency ($\Omega$), proportion of innovative cheating ($\Psi$), cost associated with mutualism ($\Lambda$) and scenario (cheaters are generalist or specialist species) for (a) a case with 10% and (b) a case with 50% of animal species cheating. The colors represent the average effect of cheating on network persistence relative to the corresponding case with no cheating ($\Omega = 0$). Positive values (blue) indicate higher persistence with than without cheating, while negative values (red) indicate the opposite, a negative effect of cheating on persistence. Results are presented for initial values of connectance $\varphi = 0.4$ and averaged over the 500 initial conditions. The black boxes represent the space in which cheating has a positive effect on network persistence, which is represented in 3D for the specialist scenario in (c) for $\Lambda = 0$ (gray) and $\Lambda = 0.3$ (dark blue). (d) Represents the volume of this 3D space as a function of the scenario and the cost associated with mutualism. Note that the full parameter space presented in (c) and (d) has a volume of 1, so the absolute volume and the fraction of volume in which cheating have a positive effect are the same measure and that a cost of $\Lambda = 0.15$ or $\Lambda = 0.3$ corresponds to 10% or 20% of benefits associated with mutualism, respectively ($\Lambda/\alpha = 0.1$ or 0.2). The data underlying this figure can be found in https://doi.org/10.5281/zenodo.10102438.

scenario of simulation, generalists versus specialists, and the cost associated with mutualism strongly affected the volume of the parameter window in which cheating had a positive effect on persistence (Fig 2C and 2D). Cheating increased network persistence by 4.7% maximum,

suggesting that if cheating could have a positive effect on persistence, it however remained a weak effect. This positive effect on network persistence tended to disappear when cheaters were generalists and with the increasing cost associated with mutualism (Fig 2B) but remained even when the proportion of cheaters was high (Fig 2D), if the overall level of cheating, measured as $\bar{\Delta} \times \Omega$, was not too high (S3 Fig).

In extreme cases, when cheaters are specialist and when cheating frequency increased above a given value, innovative cheating could have more negative effects on network persistence than conservative cheating (Figs 2A and A in S1 Results). When the frequency of innovative cheating increases, specialist cheaters switch from facilitators to competitors, because innovative cheating expands the interaction niche of specialists, thus increasing competition with other species. When performing simulation without interference among pollinators, this strong negative effects of innovative cheating at high frequency disappeared (Fig 3A). This suggests that the niche expansion of innovative cheaters that were initially specialists can promote competitive exclusion and thus decrease network persistence if cheating frequency is too high.

While these results were robust to change in connectance (Figs A and D in S1 Results), the positive effects of innovative cheating on persistence tended to vanish when performing simulations on smaller communities, with 10 species per guild (Fig 3B and 3C). This is consistent with our expectation, suggesting that a minimal number of species is needed to see positive effects of cheating appearing, which likely propagate through long indirect paths, from species to species. These positive effects of cheating are likely to be missed in a simpler model that would offer analytical solution but not model a realistic diversity of species.

Taken together, these results show that under some conditions, innovative cheating can enhance stable coexistence of species in mutualistic communities, on average. The average results presented in Fig 2A and 2B explained approximately 57% of the variance in the effect of cheating on network persistence. A part of the remaining variance was explained by the average growth rates of plants and animals and by network structure (Fig E in S1 Results). Cheating was more likely to have a positive effect on persistence when plant and animals depend weakly on each other than when this dependence is strong. On average, the effect of cheating was more positive (less negative) when initial mutualistic interaction networks were more connected, more nested, and more modular (Fig E in S1 Results), suggesting that the consequences of cheating were partially mediated by the structure of the underlying mutualistic network.

Considering the theoretical results developed above, empirical patterns of cheating should follow these conditions to maximize community persistence: (i) cheaters should be mainly specialist species; (ii) illegitimate interactions should mainly occur out of the mutualistic niche of cheaters (innovative cheating); (iii) proportion of cheaters can be high but the overall level of cheating, measured as $\bar{\Delta} \times \Omega$, should stay low (<0.5) and decrease with costs associated with mutualism; and (iv) cost associated with mutualism should not be too high.

To evaluate if these conditions are met in natural communities, we used a unique dataset of empirical interaction networks between flowering plants and pollinating birds, over 17 sites in 2 countries, Costa Rica and Ecuador. Analyzing empirical patterns using generalized linear models and type II ANOVA with Wald Chi-squared tests (*cf.* Methods), we found patterns consistent with the conditions in which cheating has a positive effect on persistence (Fig 4). First, cheaters tended to be specialist species (Fig 4B, $\chi^2 = 4.29$, *p*-value = 0.038), a result that was consistent across countries ($\chi^2 = 0.27$, *p*-value = 0.60 for the interaction). Second, birds tended to cheat mostly in an innovative way (Fig 4C), i.e., outside of the niche defined by the morphological traits of their mutualistic partners (*cf.* Methods). Third, even if the proportion of cheaters was high in some cases (Fig 4D), the overall level of cheating was low and strongly decreased with the elevation (Fig 4E, $\chi^2 = 8.94$, *p*-value = 0.003), partially because the

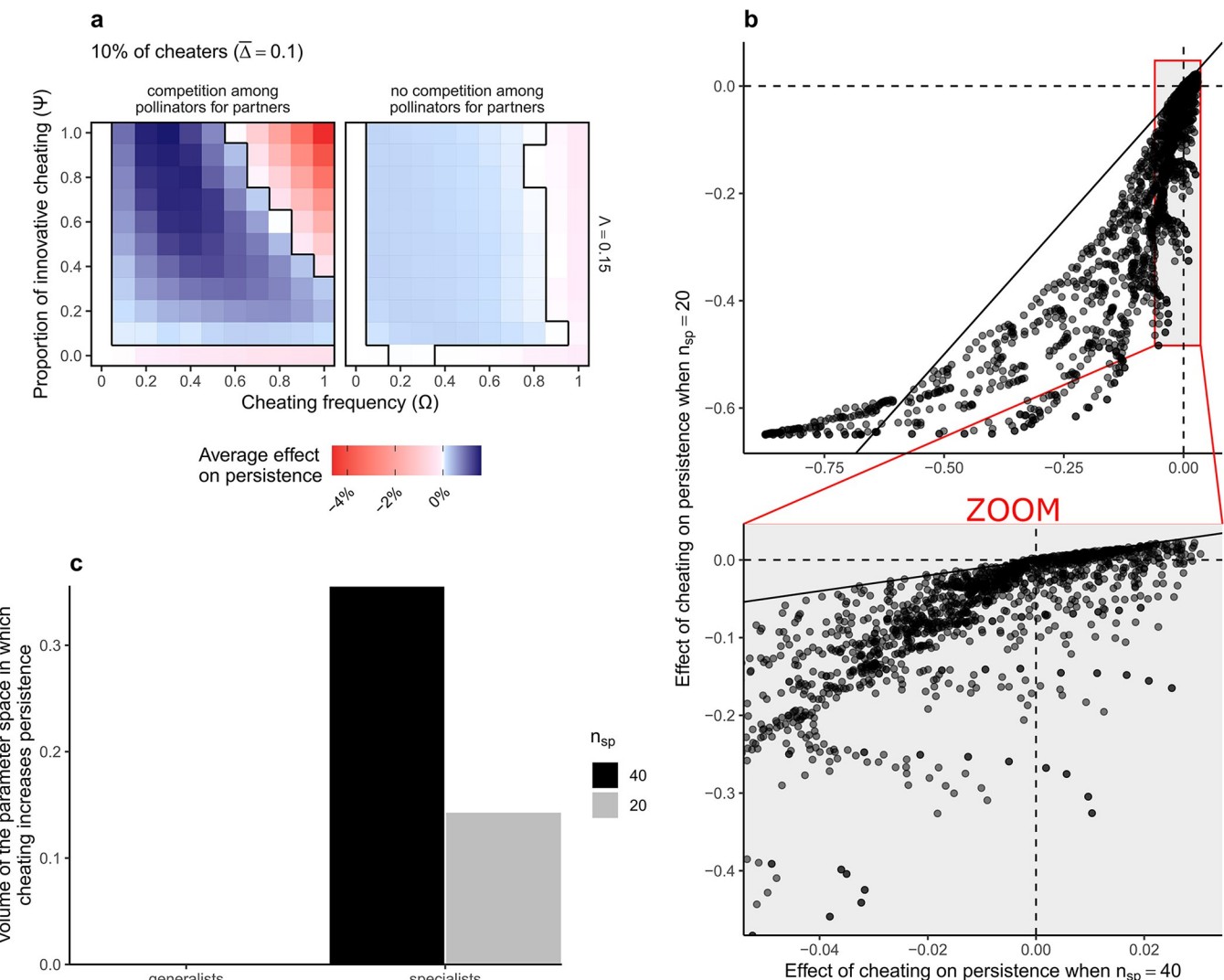

**Fig 3. Effect of cheating is mediated by interference among pollinators and diversity.** (a) Average effect of cheating on persistence, with or without competition among pollinators for partners (plants), when cheaters are specialists, for an initial value of connectance φ = 0.4. The case with competition is the same than in Fig 2A. (b) Effect of cheating on persistence in small communities, with 20 species (10 per guild) as a function effect of cheating on persistence in diverse communities, with 40 species (20 per guild). For Λ = 0.15 and φ = 0.4. The bottom part is a zoom on the gray area of the x-axis. The solid black line shows the first bisector while dashed black lines show the zero values (no effect of cheating on persistence). Each point is a simulation with the same parameter combination, excepting the number of species. (c) Volume of the 3D cheating parameter space in which cheating has a positive effect on network persistence, as a function of the scenario and the diversity, for Λ = 0.15 and φ = 0.4. Note that when cheaters are generalist species, there was no positive effects of cheating (when Λ = 0.15), so the volumes equal zero. The data underlying this figure can be found in https://doi.org/10.5281/zenodo.10102438.

proportion of cheaters also decreased with elevation (Fig 4D, $\chi^2$ = 4.68, *p*-value = 0.03). Elevation is likely a proxy of cost associated with mutualism because it increases physiological constraints on plants and birds [26,27], likely increasing the cost of nectar production for plants and flight for birds. These trends were not significantly different between countries (*cf.* Table B in S1 Results). It was not possible to evaluate the fourth condition because cost associated with mutualism was not estimable from the data. Nonetheless, the 3 first conditions were highly consistent with the theoretical patterns of cheating that maximize network persistence.

To close the loop between theory and data, we also used these empirical networks to parameterize our dynamic model in terms of species diversity per guild, distribution of interactions,

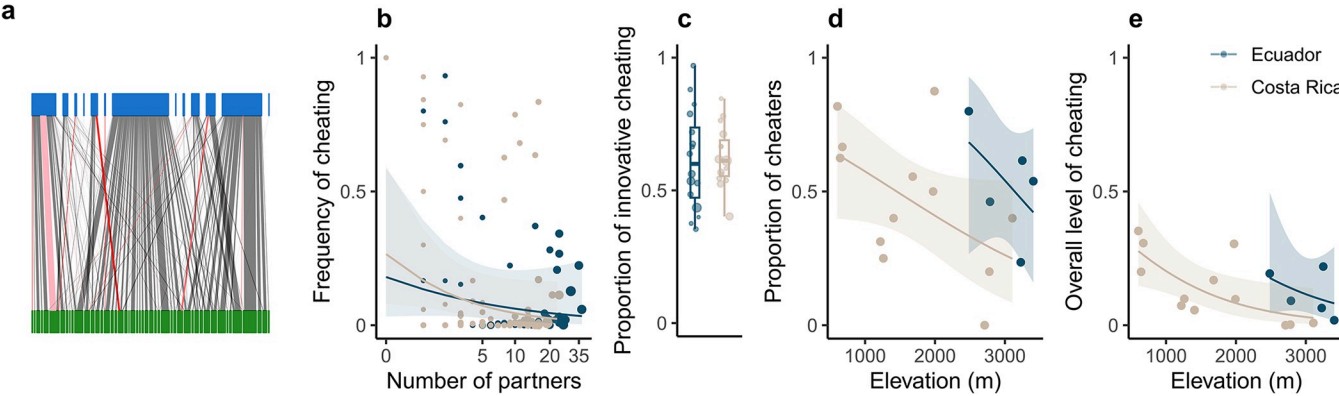

**Fig 4. Empirical cheating patterns in plant–bird interaction networks fit theoretical conditions to maximize network persistence.** (a) An empirical network of interactions between plants (green) and birds (blue) from Ecuador for a given site. Thickness of the lines is proportional to interaction frequency, while color of the lines represents cheating frequency, gray (only legitimate) and from pink (low cheating frequency) to bright red (only illegitimate). (b) Frequency of cheating as a function of mutualistic partner diversity, for each combination of bird species and site, for datasets from Costa Rica (beige) and Ecuador (dark blue). The size of the point represents the total number of interactions of the species in the given site. The solid lines show predictions of a generalized linear mixed-effects model with associated 95% confidence interval. (c) Proportions of innovative cheating, where each point is a bird species who cheated at least once. The size of the point represents the total number of interactions of the species summed over sites. (d) Proportion of cheaters and (e) overall level of cheating, for each site, as a function of elevation (in meters above sea level). The solid lines show predictions of generalized linear models with associated 95% confidence intervals. The data underlying this figure can be found in https://doi.org/10.5281/zenodo.10102438.

and cheating frequency for each plant–hummingbird pair. For parameters that were not esti- mable, we performed simulations over a gradient of values to assess the sensitivity of the results (*cf*. Methods, Fig 5). A few empirical studies have shown that costs associated with mutualism are probably low, because they do not find negative effects of robbing on seed sets [28,29] while the only study we know that quantified per-interaction benefits ($\alpha$) in pollination system found very low values [30]. When focusing on these parameter values, low costs associated with mutualism and low per-interaction benefits, we confirmed that observed cheating pat- terns increase network persistence of empirical network, relative to simulations without cheat- ing (Fig 5). Effects of cheating on network persistence in empirical networks became negative when considering high levels of per-interaction benefits and cost associated with mutualism, but this unlikely to be the case in nature. Moreover, we found that, in average, observed cheat- ing patterns tend to lead to higher persistence than randomized ones (Fig F in S1 Results), when cost associated with mutualism and per-interaction benefits were low, substantiating the fact that natural eco-evolutionary dynamics can shape stabilizing patterns of cheating.

## Discussion

We described 2 different kinds of cheating in mutualistic systems: conservative cheating aris- ing when a species profits from a mutualistic partner, and innovative cheating, when a species profits from a species that is not a mutualistic partner. Highlighting these nuances in cheating allowed us to unify in a common theoretical framework 2 different conceptions of cheating: a concept inherited from game theory in which species cheat to escape cost (conservative cheat- ing, see ref. [9,14]) and empirical observations showing that species cheat to access new resources (innovative cheating, see ref. [18]). Here, we show that these nuances in cheating are important because they have different consequences for network persistence. While conserva- tive cheating almost always negatively affects network persistence, innovative cheating, the most prevalent kind of cheating in our empirical dataset, can have a positive effect on network persistence. In our empirical study case of plant–bird interactions, simulations and data analy- ses consistently show that cheating can increase the number of species able to coexist at

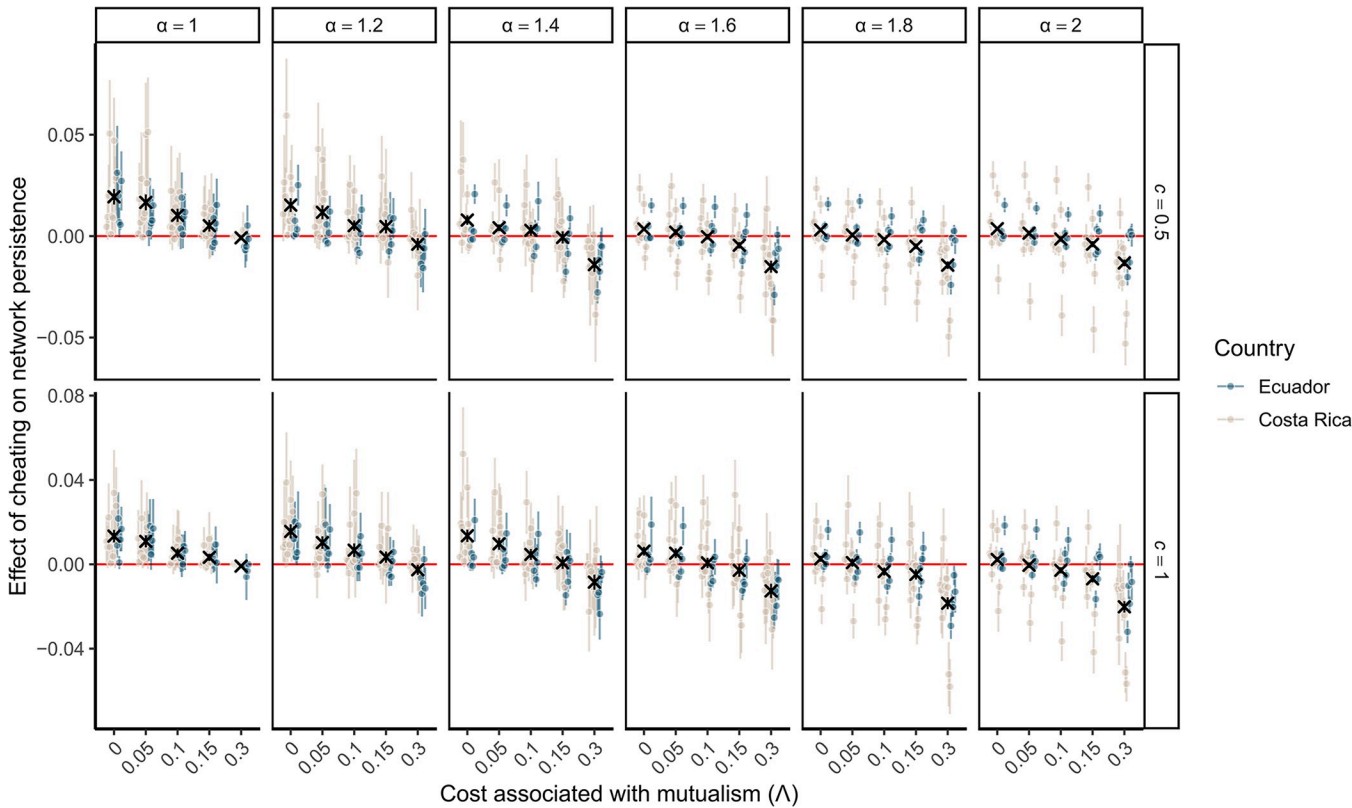

**Fig 5. Parameterizing theoretical models with empirical data reveal that cheating can increase network persistence, when cost associated with mutualism is low.** Effect of cheating on network persistence when parameterizing the model with empirical patterns of cheating, as a function of per-interaction benefit ($\alpha$), strength of competition for mutualistic partners ($c$) and cost associated with mutualism ($\Lambda$). The effect of cheating on persistence is calculated by subtracting the persistence values obtained with the simulations parameterize with all interactions (including cheating) from those obtained when only mutualistic interactions are used to parameterize the model (excluding cheating). Points represent the average effects over the 100 simulations performed over different growth rate vectors, while error bars represent the standard deviation. Black cross represents the overall average (over sites) and associated 95% confidence interval (vertical error bars, often hidden because too small). Note that the x-axis is not linear, there is a jump between $\Lambda = 0.15$ and $\Lambda = 0.3$. The data underlying this figure can be found in https://doi.org/10.5281/zenodo.10102438.

ecological equilibrium. Evolutionary theory has proposed answers to the paradox of the persistence of mutualism in presence of cheaters [9,31]. Here, we show that these cheaters are not rare in nature, and that they can enhance species coexistence in mutualistic communities instead of destabilizing them. Since cheating has been reported in many kinds of mutualisms, such as mycorrhizal [5], cleaner fish-client [32], or seed dispersal [7] interactions, it could be a common mechanism driving ecological stability of natural communities.

Most empirical studies focusing on cheating in mutualistic networks have used indirect proxies of cheating, such as marks on flower corollas left by nectar robbers [23,24] or heterotrophic status of plants in plant-mycorrhizal networks [5]. Although these proxies are valuable, they do not allow to characterize cheating as conservative or innovative, because cheating is defined at the species level. Here, by using camera traps we were able to characterize each interaction as legitimate or illegitimate, and thus define cheating at the interaction level, allowing us to estimate the relative importance of conservative and innovative cheating. Our analyses of this unique empirical dataset show a continuum gradient between cheating and mutualism: cheaters are common (up to 60% of the bird species, Fig 4D) but they often cheat at a low frequency. Using theoretical simulations, we show that such low cheating frequency can dramatically affect network persistence, negatively or positively depending on the context

(innovative versus conservative, cost and identity of cheaters). Previous work has shown that this continuum from mutualistic to antagonistic interactions, embedded in mutualistic networks, is important for community stability [7,33]. We extend these results by showing the distribution of the antagonistic links versus mutualistic ones has important consequences for stability. When considering overlapping commensalistic/antagonistic and mutualistic links (i.e., conservative cheating), network persistence is lower than when commensalistic/antagonistic and mutualistic links occur between different species pairs (i.e., innovative cheating).

Our theoretical results highlight 4 conditions that can turn cheating into a factor increasing network persistence: (i) cheaters should be mainly specialist species; (ii) illegitimate interactions should mainly occur out of the mutualistic niche of cheaters (innovative cheating); (iii) the proportion of cheaters can be high but the overall level of cheating, measured as $\bar{\Delta} \times \Omega$, should be low ($<0.5$) and decrease with costs associated with mutualism; and (iv) costs associated with mutualism should not be too high. Parameterizing our model according to these rules led to ecological equilibriums that were on average more diverse than other parameter combinations. However, since our model did not include evolution, the question remains whether these ecological equilibriums are also evolutionary stable, that is can we expect to find them in nature.

Our analyses of empirical data suggest that patterns of cheating following the 4 conditions described above can be found in nature. Empirical communities appeared to be in the parameter window in which cheating tends to optimize ecological stability. Strong cheaters were specialist species, cheaters mainly cheated in an innovative way and the overall level of cheating was relatively low and decreased when costs associated with mutualism are likely to be greater (i.e., higher elevations). However, the cost associated with mutualism is a key parameter that determines the outcome of cheating on community stability, but we were not able to assess it from empirical data. Assessing this cost would require complex physiological measurement on birds and plants that currently have been only rarely done [34,35] and never on both guilds simultaneously. Although these missing measures of cost associated with mutualism add a layer of uncertainty on the effect of cheating on community stability, our theoretical results show that even with a cost reaching 20% of the mutualism benefits, innovative cheating can still have a stabilizing effect, in a very narrow parameter combination (Figs 2B and Fig A).

The empirical patterns observed are the result of an interplay between community assembly processes and evolutionary dynamics [36], suggesting that, against what we could have expected, eco-evolutionary dynamics have shaped cheating patterns in a way that it increases community stability. The selection for innovative cheating in specialist species could be explained by the ability of species to select partners, which could lead to a competitive disadvantage for cheaters, strong enough to prevent generalists to become conservative cheaters, but still allowing specialist species to evolve as innovative cheaters. It has been shown recently that the coexistence of cheaters and mutualists can be evolutionary stabilized because of the ability of species to select the partners to which they give benefits [10]. This ability to choose can be itself evolutionary stabilized by the presence or emergence of cheaters at each generation [37], thus leading to a loop of eco-evolutionary feedbacks that could explain the empirical patterns observed here. However, it is hard to substantiate this idea because studies focusing on cheating and evolution of mutualism have only considered conservative cheating [8–10,31,37]. Innovative cheating could also emerge because of divergent optimal trait matching between plant and pollinators [38], leading to an arm race between them, than is likely to evolve towards antagonistic interactions [39], and thus innovative cheating in this context. Given the importance of innovative cheating for community stability, our results stress the need to revisit eco-evolutionary theory of cheating in a network context [11,40]. This future

direction would provide new insights on the astonishing faculty of evolutionary processes to sometimes create structures that promote ecological stability at the community level [41].

Finally, since our theoretical model went through a necessary step of oversimplifying natural systems, we likely neglected subtleties in the way species cheat that could affect the consequences of cheating for community stability, for example, behavioral changes of pollinators due to robbers [42]. It has been shown that many opportunistic pollinators use already existing holes in flower corolla [17,43]. In this case, there is likely a difference of cost for plants between the primary cheating event, degrading the floral parts that involve an additional cost for plants, and secondary cheating events that only involve the cost associated with mutualism (i.e., nectar production). In addition, consequences of such flower degradation likely depends to when it occurs, after or before pollination, suggesting that the sequential order of these mutualistic/antagonistic interactions is likely to drive the overall effect of cheating on lifetime fitness, as suggested in plant–ant interactions [44]. However, our results offer a new perspective on cheating in a mutualism context, extending the classic view inherited from the game theory, to describe a kind of cheating, innovative cheating, that was previously neglected in theoretical works. We derived a set of simple conditions turning cheating in a mechanism enhancing species coexistence, showing that antagonistic interactions embedded in mutualistic networks can play a key role in community stability. While previous studies on evolution have shown that mutualism and cheaters can coexist in the same communities [8,9], here we show that this mixture can increase the number of species coexisting in nature.

## Materials and methods

### Theoretical model

We used a generalized Lotka–Volterra model with 2 guilds, plant ($P_i$) and pollinator ($A_j$), modeling temporal variations of abundance for each species. We incorporated 4 parameters ($\bar{\Delta}$, $\Omega$, $\Psi$, $\Lambda$) that control respectively for the proportion of species cheating, proportion of links affected by cheating for each cheater (cheating frequency), the proportion of innovative cheating (versus conservative cheating), and the cost associated with mutualism. Since we expect that effect of cheating can be partially mediated by competition for partners, we chose to use a complex functional response, accounting for interference among species from the same guild (cf. Eqs (1) and (2)).

We generated a Gaussian morphological trait for each species to define an initial binary interaction matrix ($I_M$) with a given connectance (cf. S1 Methods). $I_{M_{ij}}$ was the pairwise interaction strength between plant $I$ and pollinator $j$, and equaled either zero (no interaction) or one (mutualistic interaction). $I_{M·j}$ defined the mutualistic niche of pollinator $j$, i.e., all partners with which pollinator $j$ interacted mutualistically. The mutualistic niche of plant $i$ was given by $I_{M_i}$.

Each species of pollinator was defined by a binary state, cheaters (1) versus non cheater (0), stored in the vector $\Delta$. The number of cheaters was defined as $round(\bar{\Delta} n_A)$, $\bar{\Delta}$ being the proportion of species cheating. To define cheaters, we used 2 different scenarios, one in each of those cheaters are the most specialist species and one in which they are the most generalist species (cf. S1 Methods). Each cheater cheated with a frequency $\Omega$ of their initial legitimate interactions. This amount of cheating was divided in conservative cheating $(1-\Psi)\Omega$ and innovative cheating $\Psi\Omega$. The cost associated with mutualism was set by parameter $\Lambda$, which could vary between 0 and $\alpha$, the per-interaction benefit of mutualistic interactions, to be in a mutualism case.

Thus, a plant $i$ interacted in a legitimate way with pollinator $j$ with a strength $M_{ij} = I_{M_{ij}}(1 - \Delta_j\Omega)$. The same pair of species interacted illegitimately (with benefits only for

the pollinator) with a strength $C_{ij} = \Delta_j \Omega((1 - \Psi)I_{M_{ij}} + \Psi(1 - I_{M_{ij}}))$, where $\Delta_j \Omega(1 - \Psi)I_{M_{ij}}$ was the conservative cheating and $\Delta_j \Omega \Psi(1 - I_{M_{ij}})$ the innovative cheating.

The model was described by the following system of differential equations for the plants and the pollinators, respectively:

$$\frac{dP_i}{dt} = P_i \left( r_{P_i} + \frac{(\alpha - \Lambda \sum_{j=1}^{n_A} M_{ij}A_j)}{1 + \beta \sum_{j=1}^{n_A} M_{ij}A_j + c \sum_{k=1}^{n_P} \prod_{ik} \times P_k} - \frac{\Lambda \sum_{j=1}^{n_A} C_{ij}A_j}{1 + \beta \sum_{j=1}^{n_A} C_{ij}A_j + c \sum_{k=1}^{n_P} \Upsilon_{ik} \times P_k} - \sum_{k=1}^{n_P} c_{P_{ik}} \times P_k \right) \quad (1)$$

$$\frac{dA_j}{dt} = A_j \left( r_{A_j} + \frac{\sum_{i=1}^{n_P}[\alpha(M_{ij} + C_{ij}) - \Lambda M_{ij}]P_i}{1 + \beta \sum_{i=1}^{n_P}(M_{ij} + C_{ij})P_i + c \sum_{k=1}^{n_A} \Theta_{jk} \times A_k} - \sum_{k=1}^{n_A} c_{A_{jk}} \times A_k \right) \quad (2)$$

The between parentheses part of Eq (1) can be divided into 4 parts.

First, $r_{P_i}$ was the basal growth rate of the focal species. Growth rates were scaled between −0.5 and −0.001, and drawn randomly from a beta distribution, to sample a wide variety of possible vectors of growth rates (cf. S1 Methods).

The second part modeled the balance between benefits and cost associated with mutualistic interactions. For plants, per-interaction benefit ($\alpha$) is the link between visits and the number of successful pollination events, while the cost associated with mutualism ($\Lambda$) is the cost of nectar production on survival. To have mutualistic interaction, the benefits in terms of reproduction should be higher than the cost in term of survival, so we fixed $\alpha > \Lambda$ (Table 1). Thus, overall benefits increased with the product of legitimate interaction strength ($M_{ij}$) and abundance of the interaction partners by a rate $\alpha - \Lambda$. But since successful pollination cannot increase infinitely with the number of visits because of physiological limits, benefits saturated with a rate $\beta$, which can be represented by the time a pollinator need to visit and access a flower of the focal species (handling time, Table 1). Benefits also saturated because of competition among plants for shared pollinators, by a rate $c$, $\prod_{ik}$ being a vector of interaction similarity between $P_i$ and $P_k$ (cf. S1 Methods).

The third part modeled the cost of cheating interactions only, regardless of their kind, innovative or conservative. For the costs of illegitimate interactions for plants, we used a similar functional response than for mutualistic interactions. Similarly, costs of illegitimate interactions increased and saturated with the product of the amount of cheating in interactions $C_{ij}$ and abundance of the interaction partners by a rate $\Lambda$ and $\beta$, respectively. Costs also saturated because of dilution of cheaters among plants sharing cheaters ($\gamma_{ik}$, cf. S1 Methods), by a rate $c$.

Finally, the fourth part modeled the competition among plants, independent from mutualistic partners, for example, competition for space. For simplicity, here we considered only intraspecific competition setting interspecific competition strength for space at zero (Table 1).

Eq (2) can be divided in analogous parts, except that the third part described in Eq (1) is absent, as the cost-benefit balance associated with mutualism and cheating could be grouped in 1 functional response, because all interactions with plants, regardless of their legitimate or illegitimate status, participated to competition among pollinators. Since changing $\beta$ or $\alpha$ roughly leads to the same result, for simplicity we fixed $\beta = 1$ for all species. Then, we set the value of $\alpha$ in a way that species can theoretically persist when there is no cheating, but not too high to be able to observe variation in persistence. Thus, we set $\alpha = 1.5$ for all species (cf. S1 Methods).

## Numerical simulations

We created 500 different set of species, with different traits and growth rates, while fixing $n_A = n_P = 20$. For each of these initial conditions we solved the model numerically, for the 3,630

**Table 1.  Parameter values and initial values for simulations.**

| Parameter or variable | Meaning | Value | Vary with parameter combination | Vary among initial conditions |
|---|---|---|---|---|
| $P_{i_{t=0}}$ | Initial abundance of plant species $i$ | 1 | No | No |
| $A_{j_{t=0}}$ | Initial abundance of pollinator species $j$ | 1 | No | No |
| $\alpha$ | Per-mutualistic interaction benefit | 1.5 | No | No |
| $\beta$ | Per-interaction handling time | 1 | No | No |
| $c$ | Per-capita interference for partners, within guild | 1 (or 0 for pollinators in SM) | No | No |
| $r_{Pi}$ | Plant basal growth rates for species $i$ | $\epsilon$ [−0.5, −0.001], $cf$. SM | No | Yes |
| $r_{Aj}$ | Pollinator basal growth rates for species $j$ | $\epsilon$ [−0.5, −0.001], $cf$. SM | No | Yes |
| $c_{Pik_{k\neq i}}$ $c_{Ajk_{k\neq j}}$ | Strength of interspecific competition strength for space, within guilds | 0 | No | No |
| $c_{Pik_{k=i}}$ $c_{Ajk_{k=j}}$ | Strength of intraspecific competition for space | 1 | No | No |
| $n_P$ | Number of plant species | 20 (or 10 in SM) | No | No |
| $n_A$ | Number of pollinator species | 20 (or 10 in SM) | No | No |
| $I_{Mij}$ | Per-capita interaction strength between plant $i$ and pollinator $j$ | 0/1, $cf$. SM | No | Yes |
| $\bar{\Delta}$ | Proportion of species cheating in pollinators | 0.1 -> 1 per 0.1 | Yes | No |
| $\Omega$ | Frequency of cheating for cheaters | 0 -> 1 per 0.1 | Yes | No |
| $\Psi$ | Proportion of innovative cheating | 0 -> 1 per 0.1 | Yes | No |
| $\Lambda$ | Per-interaction cost associated with mutualism | 0/0.15/0.3 | Yes | No |
| $\varphi$ | Initial connectance of $I_M$ | 0.2/0.3/0.4 | No | No |

SM = S1 Methods

chosen possible combinations of cheating parameters ($\Delta$, $\Omega$, $\Psi$, $\Lambda$, Table 1). For each of these combinations and initial conditions we applied 2 scenarios, one in which cheaters were the generalist pollinator species, another one in which they were the most specialist species ($cf$. S1 Methods). We also performed all these simulations with 3 different values of initial connectance ($cf$. Table 1). That led to $500 \times 3{,}630 \times 2 \times 3 = 10{,}890{,}000$ simulations. We stopped a simulation when the variance of species abundance over the last 10 time steps was lower than $10^{-14}$ or when the number of iteration reached 8,000. In >99.98% of the simulations, this was enough to reach a stable steady equilibrium (all eigen values of the Jacobian matrix have negative real parts).

We also performed additional simulations to explore side mechanisms, using a given value of connectance ($\varphi = 0.4$) and cost associated with mutualism ($\Lambda = 0.15$). To test if cheating can still have a positive effect in smaller communities, we performed simulations with $n_A = n_P = 10$. To investigate the role of competitive exclusion in the fact that innovative cheating strongly affects persistence in a negative way when cheating frequency is close to one, we performed simulations without competition (i.e., interference) among pollinators for shared partners.

## Networks metrics

To understand how the initial structure of the network affects persistence, we measured the nestedness and the modularity of the initial binary interaction matrix ($I_M$), using NODF index [45] and Newman's modularity measure [46], respectively. The Newman's modularity measure is the likelihood-equivalent of the final proposed module structure after optimization,

which was done using the algorithm proposed by Beckett (2016) [47] as implemented in R package *bipartite* [48].

We evaluated network stability at equilibrium, as network persistence. Network persistence was measured as the percentage of persisting species at equilibrium, with an abundance greater than $10^{-5}$ at the equilibrium, other species being considered as extinct, for each community and simulation.

## Analyses of theoretical results

We presented theoretical results either as the absolute values of network persistence over parameters and scenario values (Fig A for example), or as the difference between the case without cheating and the focal simulation for similar initial conditions (Fig 2). The later one directly represents the effect of cheating on network persistence. These absolute or difference measures were averaged over the 500 initial conditions.

To determine the proportion of variance explained in cheating effect on persistence by our parameter and scenario combinations (group), we calculated the overall variance of network persistence over all simulations ($V$), and the variance within group ($V_g$), corresponding to the variance of centered persistence values for each group (minus the average persistence of that group). These centered values represent the residual effects of cheating on persistence, once we have accounted for parameter and scenario combinations. Then, we estimated the proportion of variance explained in cheating effect on persistence by our parameter and scenario combinations as 1- $V_g/V$.

To understand what drives the residual cheating effect on persistence, we used a linear model to explain the residual effect of cheating by: average growth rates of animals and plants; and connectance, modularity, and nestedness of initial mutualistic interaction matrix. To break the correlation between connectance and nestedness and between connectance and modularity, we centered values of nestedness and modularity by the average nestedness and modularity, respectively, for each of the connectance values.

## Empirical dataset

To assess the conservative/innovative character of cheating in empirical communities, we used data of interactions among flowering plants and birds, hummingbirds (Trochilidae) in Costa Rica and hummingbirds and flowerpiercers (*Diglossa* sp.) in Ecuador. Sampled sites were located over an elevation gradient in the tropical forests of Costa Rica and Ecuador (Table B in S1 Results), in which interaction were recorded along transects using camera traps, as described in Graham and Weinstein (2018), ref. [49]. With each recorded interaction, the information of the interaction status, legitimate (bird bill entering the corolla opening) versus illegitimate (bird piercing the corolla), was recorded. Thus, here we assumed that illegitimate interactions do not bring any benefits to the plants and that when a bird bill enters the corolla, the interaction is legitimate. Previous results have shown that even interaction that seems illegitimate can participate to pollination and thus be mutualistic [50]. However, because piercing is often done at the base of the corolla, robbing interaction described here are likely to be non-efficient in terms of pollination, so as cheaters, although a strict classification would require fitness estimate [31].

Data from Costa Rica includes 18,199 interactions from 12 different sites, sampled monthly between June 2019 and May 2021. Data from Ecuador includes 15,459 interactions from 5 different sites, sampled monthly between February 2017 and December 2019. To obtain interaction frequency independent from the sampling pressure, each interaction count was divided by the number of hours of sampling on the corresponding plant, and then multiplied per 24 to

get interaction frequencies in interaction.day$^{-1}$, for legitimate and illegitimate interactions. Then, in both datasets, we aggregated interactions over time to get 1 interaction network per site (Figs G and H in S1 Results). Since on the transect plant abundance have been recorded independently, we were able to compare the diversity of flowering plants we sample relative to the observed diversity on the transect. On average, we sampled 74% of the flowering plants present on each site, with values going from 57% to 93%. Most of non-sampled plants were either non-tubular, so unlikely to be visited by hummingbirds or species of low abundance (Fig I in S1 Results).

## Estimation of empirical cheating parameters

To compare empirical patterns with our theoretical results, we estimated the 3 parameters related to cheating in our model: the proportion of pollinator species cheating ($\bar{\Delta}$), cheating frequency of these cheaters ($\Omega$), and the proportion of innovative cheating versus conservative cheating ($\Psi$). The estimation of these 3 parameters is detailed below.

To estimate the proportion of cheaters in each site, we calculated the proportion of species with at least 1 illegitimate interaction.

To estimate the cheating frequency for each bird species in each site, we calculated the proportion of illegitimate interactions for each species for each site, as the frequency of illegitimate interaction divided per the total frequency of interaction, both summed over all the partners of the focal bird species in the given site.

To estimate the proportion of innovative cheating, we first built a 2D mutualistic niche for each bird species. To do so, we used 2 plant traits measured on flowers (*cf.* S1 Methods), the length of the corolla and its curvature. We had to exclude 17 plant species over 246 because of missing trait values. For each bird species, we then calculated a 2D kernel density distribution of mutualistic partners, over corolla length and curvature, weighting each plant by the legitimate interaction frequency with the focal bird species. This 2D density distribution models the mutualistic niche of the bird. We then did the same manipulation, but weighting the density by the illegitimate interaction frequencies, to obtain the space in which bird are cheating. Each density distribution summed to one. The overlap between the density function, measured as the area under the minimum of both functions, corresponds to the proportion of conservative cheating. It equals to one when bird cheat only within their mutualistic niche and zero when bird cheat only outside of their mutualistic niche. To obtain the proportion of innovative cheating, we simply took one minus this overlap, because the proportion of innovative cheating equals one minus the proportion of conservative cheating.

Finally, to estimate the overall level of cheating in each site ($\bar{\Delta} \times \Omega$), we calculated the average proportion of illegitimate interactions across all species of the focal site, as the frequency of illegitimate interactions divided by the total frequency of interactions, both summed over all pairs of plant and bird species in the given site.

## Statistical analyses of empirical patterns

Since elevation negatively affects net primary production of plants [26], it is expected to be positively correlated with the cost of nectar production, especially because of temperature-mediated effects on photosynthetic activity [51]. Elevation is also known to increase flight cost for birds, especially hummingbirds [27,35]. These elements make elevation a good proxy of costs associated with mutualism. To assess how the proportion of cheaters and overall level of cheating varied over elevation, for each of these response variables we used a generalized linear model with a quasibinomial error distribution and a logit link function, where we included elevation as predictor and its interaction with country.

To assess how cheating frequency varies with the level of generalization/specialization of birds, we used a generalized linear mixed-effects model with a binomial error distribution and a logit link function. This model explains cheating frequency of each bird species in each site per the logarithm of their diversity of mutualistic partners, in interaction with the country. We used the logarithm to implement a saturation to generalism, all species with many partners being similarly generalists, while considering an important difference in term of generalism between a species having 1 partner and a species having 3 partners. We also included random site and bird effects on the intercept. Diversity of mutualistic partners was calculated as the number of plants with which at least 1 legitimate interaction was recorded, for each bird species and site.

## Numerical simulations with empirical networks

To test if the observed patterns of cheating were likely to increase network persistence, we parameterized our dynamic model with empirical patterns of cheating. First, we converted empirical interaction networks presented in Figs G and H (in S1 Results) into binary interaction matrix $I$, in which $I_{ij}$ is a binary state describe if bird $j$ and plant $i$ interacted at least once mutualistically (1) or never (0). We used a binary interaction matrix as a mutualistic backbone rather than a quantitative interaction matrix because we assume that it reflects the ability of species to interact in a mutualistic way, while interaction frequencies depend much more on abundance. Second, we estimated the proportion of cheating among those species ($C_{ij}$) by dividing the frequency of illegitimate interactions per the total number of interactions. Then, we estimated the ability of this species pair to interact in a legitimate way using $M_{ij} = I_{ij}(1-C_{ij})$. In our model, abundances were modeled a part of the interaction matrix, thus this interaction matrix should reflect only the ability to interact, and not interaction frequencies.

Parameters from the functional response could not be inferred from the data: the per-interaction benefit ($\alpha$), sometimes called mutualistic strength [52,53], the cost associated with mutualism ($\Lambda$), the handling time ($\beta$), and the strength of competition for mutualistic partners ($c$). To overcome this limitation, we performed simulations over a gradient of possible values, even if it is hard to give an empirical meaning to those parameters. We thus perform simulations from low ($\alpha = 1$) to high per-interaction benefits ($\alpha = 2$) by steps of 0.2, for 3 different values of cost associated with mutualism ($\Lambda \in [0, 0.05, 0.1, 0.15, 0.3]$) and 2 different values of competition, intermediate ($c = 0.5$) and high competition strength ($c = 1$). Since changing the value of $\beta$ has roughly the same consequences as changing $\alpha$, we decided to keep $\beta$ constant. For each of these per-interaction benefit and cost combinations, we generated 100 different combinations of growth rate vectors, using equation (S1, *cf.* S1 Methods). Since 2 sites exhibited either 0% or 0.2% of illegitimate interactions, MILL and AMIG sites, respectively, we excluded them from these simulations. For each of the 15 remaining sites, 100 initial conditions and 60 parameter combinations, we performed simulations with all empirical interactions, or without cheating interactions (setting $C_{ij} = 0$, for all $i$ and $j$).

This part led to 180,000 new simulations. To analyze these simulations, we measured persistence at equilibrium, as explained above, and we subtracted the value of the simulation set without cheating to the value of the corresponding simulation with all interactions. This allowed us to obtain the effect of cheating on network persistence, presented in Fig 5.

To assess how the specific observed cheating patterns affect persistence relative to random patterns, we also run simulations parameterized with randomized proportion of cheating among species, keeping the overall level of cheating constant. Thus, we randomized the identity of the cheaters and the proportion of innovative versus conservative cheating. Since we performed 100 randomizations for each of the 100 initial conditions for each site, we were

limited by the computational cost these simulations. We could perform them only for the parameter window that is the most likely to be find in nature (see Results for references), low per-interaction benefits ($\alpha \epsilon [1; 1.4]$) and costs associated with mutualism ($\Lambda \epsilon [0; 0.05]$), for $c \epsilon [0.5; 1]$. This added 1,200,000 new simulations. For each site and initial conditions, we calculated a z-score to compare the observed persistence value (obtained when parameterizing the model with empirical patterns) to the distribution of persistence values obtained when randomizing cheating patterns. Since z-score requires that the variable are normally distributed and that persistence is bounded between 0 and 1, before calculating z-score we *logit* transformed all persistence values, adjusting values equal to 0 and 1 to 0.01 and 0.99, respectively. Z-scores that were equals to $-\infty$ and $+\infty$, because of null standard deviations were set to $-5$ and $+5$, respectively. To have 1 measure per site, we averaged the z-scores across the 100 initial conditions.

## Supporting information

**S1 Methods. Supplementary methods.**
(PDF)

**S1 Results. Supplementary figures and tables.**
(PDF)

## Acknowledgments

We would like to thank all field assistant who worked in collecting data. In Costa Rica, we thank botanist Nelson Chavez Elizondo; field assistants José Alejandro Castro Gutérrez, Krystal Zúñiga, Fabian Monge Badilla, Greivin Serrano Salazar, and Samael Padilla; the owners and administrators of the study sites, Los Nimbulos, Los Amigos del Bosque, Villa Mills Experimental Station (ACC), Villa San Miguel, Finca Boquete (Sibu), Bosque del Tolomuco, Centro Biológico Las Quebradas (FUDEBIOL), Las Nubes (York University), Sendero Los Gigantes, Refugio de Aves Los Cusingos (CCT), Hotel Rio Magnolia and Longo Mai. In Ecuador, data collection was possible thanks to Friederike Richter, Andrés Marcayata, Cristian Poveda, and Bryan Rojas with the local assistance of Wilson Hipo, Rolando Hipo, Silvio Calderón, and Roberto Pailacho. Plant species were identified by Francisco Tobar. We are also grateful with Jocotoco Foundation, Maldonado family, Antonio Páez, Adela Espinosa, and Alaspungo community. We also would like to thank Nicolas Loeuille and Carlos Melian for their insightful feedbacks on the manuscript.

## Author Contributions

**Conceptualization:** François Duchenne, Stéphane Aubert.

**Data curation:** François Duchenne, Emanuel Brenes, María A. Maglianesi, Tatiana Santander.

**Formal analysis:** François Duchenne.

**Funding acquisition:** Catherine H. Graham.

**Investigation:** François Duchenne.

**Methodology:** François Duchenne.

**Validation:** François Duchenne, Stéphane Aubert, Elisa Barreto.

**Visualization:** François Duchenne.

**Writing – original draft:** François Duchenne.

**Writing – review & editing:** François Duchenne, Stéphane Aubert, Elisa Barreto, Emanuel Brenes, María A. Maglianesi, Tatiana Santander, Esteban A. Guevara, Catherine H. Graham.

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
