## [Editor Report · Decision Letter 0]

11 May 2023

Dear Dr Duchenne, 

Thank you for submitting your manuscript entitled "When cheating turns into a stabilizing mechanism of mutualistic networks" for consideration as a Research Article by PLOS Biology.

Your manuscript has now been evaluated by the PLOS Biology editorial staff, as well as by an academic editor with relevant expertise, and I'm writing to let you know that we would like to send your submission out for external peer review.

Once your full submission is complete, your paper will undergo a series of checks in preparation for peer review. After your manuscript has passed the checks it will be sent out for review. To provide the metadata for your submission, please Login to Editorial Manager (https://www.editorialmanager.com/pbiology) within two working days, i.e. by May 15 2023 11:59PM.

Kind regards,

Roli Roberts

Roland Roberts, PhD

Senior Editor

PLOS Biology

rroberts@plos.org

---

## [Decision Letter · Decision Letter 1]

8 Aug 2023

Dear Dr Duchenne,

Thank you for your patience while your manuscript "When cheating turns into a stabilizing mechanism of mutualistic networks" went through peer-review at PLOS Biology. Your manuscript has now been evaluated by the PLOS Biology editors, an Academic Editor with relevant expertise, and by three independent reviewers. Please accept my apologies for the time taken over the challenging summer months.

You'll see that reviewer #1 is positive, calling this “ambitious and interesting,” but says s/he couldn’t access the code, and found the empirical sections much stronger than the modelling ones. S/he wants more interpretation, and some improvement to your presentation and stats. Reviewer #2 also thinks this is important, but focusses on the modelling part, which is closer to his/her specialty. S/he wants a better Introduction to frame the hypothesis and present the existing literature correctly, and also suggests that you start the modelling with simpler communities (4 species rather than 40), that you might consider removing Figs 3 and 5, and has a number of textual and semantic suggestions. Reviewer #3 is also broadly positive, but wants you to clarify the model that you’re using, establish the advance over the literature, improve the flow, and discuss any sampling bias.

I discussed the comments with the Academic Editor, who provided the following additional advice that you might find helpful:

"Overall I found the comments by the three reviewers highly concordant: while all them agreeing with the interest of the work, they all have important concerns about the presentation and structure. The MS still needs careful work to address fundamental aspects of the biological interpretation of results. This problem probably stems in a lack of clear presentation and motivation of the modeling details, that limits establishing a clear link - later in the Discussion - between model results and interpretation. Cheating in mutualisms is a biological process full of natural history details, so it is important to establish this connection between model-results-biological interpretation; otherwise the MS would read as an intellectual exercise in a vacuum. The other problem that still persists is the inclusion of a number of marginal analyses that detract focus from the main story."

In light of the reviews, which you will find at the end of this email, we are pleased to offer you the opportunity to address the comments from the reviewers in a revision that we anticipate should not take you very long. We will then assess your revised manuscript and your response to the reviewers' comments with our Academic Editor aiming to avoid further rounds of peer-review, although might need to consult with the reviewers, depending on the nature of the revisions.

**IMPORTANT - SUBMITTING YOUR REVISION**

*Resubmission Checklist*

*Published Peer Review*

*PLOS Data Policy*

*Blot and Gel Data Policy*

Sincerely,

Roli Roberts

Roland Roberts, PhD

Senior Editor

PLOS Biology

rroberts@plos.org

REVIEWERS' COMMENTS:

Reviewer #1:

The manuscript by Duchenne et al. combines a theoretical model of a mutualist network and observations of a pollinating bird-flower system to investigate how cheating can evolve at the community scale. Their theoretical models suggest that two routes of cheating evolution (which they term 'conservative' and 'innovative') could be expected to have differential effects on network persistence depending on the makeup of the original network. They then suggest that this would have consequences for which cheating mechanisms are more frequently observed in real systems. They find that hummingbird networks appear to match their predictions. 

This paper is highly ambitious and investigates an interesting topic - the potential positive effects of cheating as an example of how the impact of processes can change across scales. It contains lots of interesting novelty, is well written and appears to be well carried out, although I could not access the raw code and data. That said, in my opinion the empirical part of the paper with its clear hypotheses is considerably stronger than the modelling part, where I somewhat struggled to discern the objectives (and this is speaking as modeller!). Nonetheless, the two parts are nicely tied together. Even if there might be some scepticism about the detail of the models, the empirical results stand as something interesting and worth of explanation.

The concluding claim of the abstract that cheating is stabilising is quite a leap and lots of different definitions of stabilising being melded here. Some more description of the circumstances and mechanisms in which mutualisms aid persistence would be good. By this is I don't mean the parameters used in the model, but what it was about the model (and the empirical system) that led to these results (e.g. why destabilising effects of run-away mutualism are avoided). 

Specific points

39- I found the labels of cheating types in abstract not particularly clear. The second para of intro did a much better job and it was only then that I really understood what was intended. The first sentence of the discussion is good too. 

55- grammar?

115 - Appreciate the structure makes modelling papers a challenge, but I would have liked to see some more details of the model form before diving into the results. Linked to my general point, it is hard to shake the feeling that these results are dependent on something about the modelling approach and definition of stability. For example, was it just an effect of connectence? 'conservative' cheating leads to loss of connectence, while innovative adds connectence? 

Linked to this, the 11 million parameter combinations tested seems excessive, but is still ultimately one model form and approach. 

Why was cheating relatively rare? If defined as avoiding costs, why are most of the interactions non-cheating, costly interactions. 

135 - 'on average'?

138 - 'buffer'

186 - using symbols in this key hypothesis sentence is a bit alienating. 

Statistics and interpretation of Fig 4 - 

a) It was hard to know what to take from the image here. An interaction matrix representation may be more clear, but if idea is just to be like 'we made some networks' it should be fine (if a little pointless). Sorting the species to reduce overlap would help a lot too. 

b) Very hard to trace out what the statistical tests behind this actually are. 'First, cheaters tended to be specialist species' suggests a binary response, but the figure is continuous response( frequency of cheating). Was this a chi-squared test of including log-number of partners? 

Why the log-transformation? Not a criticism, but good to know if the result is depend on the transformation - 0-35 isn't an enormous spread. Maybe also specify in the graph axis that it is mutualistic partners?

The grey-shading here suggests a positive relationship lies within the 95% confidence interval for at least one of the countries but it is hard to read. Need to spell out a little more precisely what tests were done here rather than relying on the SI table. Was there a strong correlation in the predictors? 

Lastly, as much as I can believe the results are statistically significant, it would be good to have some sense of how much of the variation in the data is explained by the key driver in each case. Is this something that is only just detectable, or something that really controls the structure of the mutualistic networks.

c) A bit more outline of the method used to determine the mode of cheating from traits would be very helpful. It looks a nice method and could be emphasised some more. 

215 - I appreciate the objective of 'closing the loop' here, but as it stands I found it quite hard to follow figure 5 here, and I'm not too convinced how tight a match the model would be to the real system. 

418- These are big impressive numbers, but it would be still be good to give some sense of network completeness here or how robust the results are to bootstrapping the observations. As this dataset has been used repeatedly, I assume this must exist somewhere already so shouldn't require much further work. 

S7 & S8 are near enough useless - too low resolution to read lines, no species identities, and species are not sorted to minimise cross-over. 

Are the locations presented in Table S2 really millimetre accurate? I'm not clear what is going on here, but has some transformation been applied? It might be worth checking. 

Reviewer #2:

Summary: Duchenne et al.'s manuscript explores the impact of inter-specific cheating on multi-species communities, with a focus on communities of plants and pollinators . The manuscript consists of two 'chapters':

 ◦ In the first chapter, the authors develop a modified Lotka-Volterra model and then conduct simulations exploring the effect of different types of cheating (conservative and innovative) on community coexistence across various parameter combinations. The authors argue that these different types of cheating have different qualitative impacts on community coexistence, with innovative cheating maintaining diversity under certain conditions.

 ◦ The second chapter analyzes an empirical dataset of observed pollination events involving diverse flowering plants and birds from Costa Rica and Ecuador. The authors claim that the observed cheating patterns in this dataset align with the conditions predicted by the model to promote community diversity via innovative cheating.

This question of whether inter-specific cheating can contribute to maintaining diversity in complex communities is of interest to a wide variety of researchers including community ecologist, evolutionary ecologist and behavioral ecologist. To my knowledge the distinction made in this paper between conservative and innovative cheating is novel, and I found the justification for this distinction based on empirically observed plant-pollinator interactions to be compelling. Because the specific analysis of plant-pollinator interactions is outside of my area of expertise, for this review I will primarily focus on the introduction and theoretical portion of the manuscript. This portion of the manuscript includes some interesting ideas and preliminary results, but I have several major and minor comments which I hope will prove useful and that would need to be addressed in a major revision.

Major Comments

 1. The introduction fails to clearly justify the main hypotheses and does not embeds the manuscript within the broader literature on the ecology and evolution of mutualism. Relevant papers exploring cheating's impact on diversity and coexistence are ignored including Yoder and Nuismer 2010, Leinweber, Inglis and Kummerli 2017, and Heath and Stinchcombe 2014. The authors do not need to be systematic, but considering a wider body of work may help them clarify several claims in the introduction which are overly simplistic and/or unsubstantiated, for example: i) invasion of conservative cheaters always leading to the exclusion of mutualistic partners (lines 71-72), ii) innovative cheating leading to competitive exclusion of other cheaters (lines 74-76), and iii) innovative cheating increases coexistence for specialists but decreasing it for generalists (lines 76-82).

 2. The results section starts with simulations involving a complex ecosystem of 40 species, which makes it challenging to determine the precise mechanism through which innovative cheating is promoting coexistence and how it might depends on plant and/or pollinator diversity. Starting with a minimal example (e.g., 2 pollinators and 2 plants) would greatly strengthen the manuscript, providing clearer illustration of the mechanism at play and potentially being amenable to analytical solutions. 

 3. The motivation for Figure 3 (lines 189-203) is unclear and appears disconnected from the main question of the manuscript. To enhance clarity, I would urge the authors to either drop this section or work to better incorporate this result into the manuscript by comparing the effects of different network metrics with the network metrics observed in the empirical data.

 4. The results shown in Figure 5 (lines 215-221) are poorly motivated and not especially convincing. If I understand correctly, the authors are trying to show that when they constrain the model to reflect parameters that can be estimated, the 'un-estimatable' parameters' will alter coexistence in a characteristic manner and tend to have a positive impact on coexistence in the region we expect 'in nature' (line 221).This argument is both convoluted and not particularly convincing given that i) the authors acknowledge the parameters 'have no empirical meaning' (lines 483-484) ii) there are several big assumption made in the parameter estimation (for example the authors are not paramaterising the model using the actual interaction estimates (Figure S7-S8), but only the aggregated statistics, Lines 470-479) and iii) the authors don't actually explain why low per-interaction benefit and costs are expected 'in nature' (line 221). The authors may want to consider removing this section or placing it in the supplement and instead spending more time in the discussion reviewing the actual empirical evidence for a low cost of mutualism 'in nature' (which was already established as important in figure 2).

 5. The methods section (lines 314-374) describing the model and simulations requires substantive revision and clarification as it is very hard to follow. A diagram illustrating the parameter sampling and simulation procedure, along with plots showing the impact of different parameters on the interaction matrix, would be extremely helpful. More-over several modeling choices, seems unnecessarily complicated given the fairly simple question the authors are asking. For example I) why go through the steps of building the binary interaction matrix from a sampling of species traits (lines 319- 325) ii) why sample growth rates from a beta distribution with shape parameter sampled from another distribution (lines 342- 344 and supplementary methods) and iii) why choose such a complex functional response (supplementary method). It is unclear if the authors would obtain similar results if they had adopted a simpler approach by say a) sampling a binary interaction matrix directly, b) sampling growth rates from a single distribution and c) using a more simple functional response. If they would not obtain qualitatively similar results then I would be very concerned about the claims they are making given that these arbitrary modeling choice do not naturally emerge from the empirical data. 

Minor Comments

 1. General: The authors repeatedly refer to 'illegitimate' and 'legitimate' interactions, a vocabulary that I found to be unnecessarily confusing. Wouldn't 'cooperative interaction' and 'cheating interaction' suffice?

 2. General: On a related note the terms 'network persistance' and 'stability' and are used rather loosely throughout the manuscript and could be dropped for the sake of precision and clarity. In all the simulations the authors are specifically talking about coexistence and diversity (i.e % species coexisting at equilibrium), and this should be reflected in the text.

 3. Lines 50-57: The introduction and abstract would be strengthened by discussing mutualism in a broader phylogenetic context (i.e not just plants and pollinators)

 4. Lines 90-91: - The authors need to be much more explicit here and in the abstract about what aspects of the empirical dataset are new for the paper and what parts have been published in previous work. I have gone over the methods multiple times and it is still unclear to me if the authors are analyzing completely new camera trapping data (which would be extremely impressive), whether they are re-classifying interactions recorded in previously collected videos as legitimate or illegitimate (still impressive) or whether they are simply re-analyzing previously collected data.

 5. Figure 2D: - The authors should also show the volume of parameters space in which cheating decreases 'persistence', given that the claim is about net effect.

 6. Line 153-159: This paragraph is very speculative with respect to actual ecological mechanisms at play. But the authors have the full model and so should be able to describe precisely what is going on in their simulations. For example could they not test whether there is competitive exclusion by cheaters due to niche overlap by looking at the effect on pollinator diversity versus plant diversity (rather than simply looking at overall species diversity at equilibrium). If the mechanism in lines 157-159 is true, naively I would expect a greater drop in pollinator diversity compared to plant diversity.

 7. Lines 160-174: As stated in my major comments I think this entire section could be removed (or moved to a supplement) and the message of manuscript would be completely unaltered. If the authors choose to keep this section then it needs to be expanded upon because currently there is insufficient detail to understand what the authors are claiming/doing. For example where does the 51% number come from, why does the correlation of residuals to growth rate imply cheating has a larger impact when interaction are weak and why would one a-priori expect different network topologies to differentially modify the impact of innovative cheating.

 8. Lines 215- 221 See previous comment.

 9. Figure 5: Is there any reason why the x axis in figure 5 is non-linear? It should be pretty easy to add simulations for 0.2 and 0.25. 

 10. Figure 5: The authors do not discuss in the main text why they are performing simulations over two different 'strength of competition' (c). If the authors choose to keep this figure, I would only show the results for one set of simulations in the main figure for the sake of clarity (and move the other set of simulations to the supplement)

 11. Lines 238-239: 'always' seems a bit strong given that there is a region in figure 2D where conservative cheating has a positive effect.

 12. Lines 245-250: The authors could emphasize this point more (especially in the introduction). I found the connections to the rich empirical dataset to be the biggest strength of this paper and wish the authors had done more to leverage the data shown in figure S7 and S8 .

 13. Line 295: Citation needed

 14. Line 300-302 the lack of additional cost to the plant of cheating is a major assumption of the model (line 111-112) and needs more justification/discussion. How might this additional cost impact the main theoretical predictions in Figure 2. Is it possible to get some idea of whether innovative cheating interaction have additional cost from the camera trap data (damage to flower etc).

 15. Lines 314-387. The authors should make sure they define every parameter before or when they are referencing it (for example in Line 332 I don't think alpha has been defined). Similarly the authors should also make sure they do219 persistence in empirical networks became negative when considering high levels of per-interaction benefits not define parameters or objects that are not strictly needed to explain the model and/or simulations (I.e the second matrix in line 324-325).

 16. Lines 363-374 The authors should clarify what coding language the model was written in, what packages were used (and the version of all relevant software).

 17. Table S1 : Every parameter value not varied in the simulations needs to be justified or mentioned at some point in the methods or main text. 

 18. Lines 475-479. I struggled to understand this point, could the authors not just have converted the quantified interaction networks shown in Figure S7-S8 into a binary matrix based on presence/absence of an observed interaction? 

Reviewer #3: 

Duchenne et al., do a good job of exploring the impact of cheating in mutualistic networks using a combination of Lotka-Volterra population models and bipartite network representations of plant-pollinator interactions. Overall, the manuscript is well-written and effectively explained in most areas, but I would suggest some revisions and additions before considering it for acceptance.

MAJOR SUGGESTIONS

One important improvement would be to provide more information about the authors' modeling framework in the introduction section, before delving into the results. Clarifying how their approach differs from previous works will help readers better understand the significance of their research. Additionally, in the results section, adding more details about the model would enhance the reader's comprehension.

Furthermore, the Methods section could benefit from a bit more elaboration. Currently, it may appear dense to a first-time reader, so enhancing its flow within the text would be advantageous. While the provided table is useful, some information may require going back and forth between sections, so integrating pertinent details (including biological insights) directly into the text would improve the overall accessibility of the content. 

One notable concern revolves around the assumption of the cost of mutualism and its applicability in nature. While such an assumption is necessary for any modeling framework, it could have secondary repercussions on the results. The authors should comment on the potential robustness of their findings to this assumption and discuss its implications on the overall conclusions.

Furthermore, providing more information about the data and sampling completeness would be valuable. Readers would appreciate insights into whether the data collection time and style introduced any biases in recording interactions. This information will strengthen the reliability and validity of the study's outcomes.

Overall, addressing these major suggestions will enhance the manuscript's clarity and strengthen its overall merit for potential acceptance.

MINOR COMMENTS AND QUESTIONS

Line 35: [Minor] I was confused initially (before reading the MS) about what methods you are actually using - maybe replace the 'theoretical model' with a couple of phrases about the combination of network methods and the generalized Lotka-Volterra model.

Line 53: [Minor] Maybe cleaner fish-client fish (interactions) or fish cleaning (interactions)?

Line 57: (studies) has -> have

Line 57: (consequences) has -> have

Line 71: [Minor] define persistence for the readers in this specific context

Line 86: [Minor] go ahead and describe your theoretical model in a little more details (even from the beginning of the paper, the readers have no idea what you mean by your model - and what kind of model)

Line 108: [Major] Is the cost of mutualism the same for both partners? What kind of cost is it? I assume for simplicity of modeling, you assumed it to be a constant across all species. Exact reciprocity of the cost, I feel, is non-biological. From an energetic perspective, the cheaters are still using energy to gain plant resources, right? So, the cheaters should just have a lower energetic cost but not zero and also the pollinators should have a different cost associated than plants.

Line 112: due to cheating? And remove previous due?

Line 116: [Minor] The reader still doesn't know how your Lotka-Volterra model is coupled with your network; also you use proportion in all your plots and not percentage for persistence

Line 198-200: [Minor] Please provide citations

Line 324: you mean Ioij instead of Imij for the first expression?

Line 346: Are benefits alpha just some proportionality constant related to successful pollination? How do you think about it biologically?

Line 348: Is beta just handling time (as in table) or has a different meaning as well? Also, just mention it in the text what the parameters mean succinctly.

---

## [Decision Letter · Decision Letter 2]

1 Nov 2023

Dear Dr Duchenne,

Thank you for your patience while we considered your revised manuscript "When cheating turns into a stabilizing mechanism of mutualistic networks" for publication as a Research Article at PLOS Biology. This revised version of your manuscript has been evaluated by the PLOS Biology editors, the Academic Editor and the original reviewers.

Based on the reviews, we are likely to accept this manuscript for publication, provided you satisfactorily address the remaining points raised by the reviewers and the following data and other policy-related requests.

IMPORTANT - Please attend to the following:

a) Please make your title more specific; we suggest "When cheating turns into a stabilizing mechanism of mutualistic networks in ecological communities" or "When cheating turns into a stabilizing mechanism of mutualistic networks in plant-pollinator communities"

b) Please address the remaining concerns raised by reviewers #1 and #2. In case it's helpful, the Academic Editor said "In particular I agree with rev #2 that evidence for competition per se to be a driver of improved coexistence might be strengthened and much better supported by a contrast against a null model. Just generating multiple random networks and comparing the empirical data against the distribution of random values will be simple to achieve and represent a true added value to this already interesting MS. This type of contrast of network-derived, empirical metrics with random-generated ones is a very frequent procedure in complex network analyses."

c) Please address my Data Policy requests below; specifically, we need you to supply the numerical values underlying Figs 2ABCD, 3ABC, 4ABCDE 5, S1AB, S2, S3, S4AB, S5ABC, S6, S7, S8, either as a supplementary data file or as a permanent DOI’d deposition. I note that you already have an associated Zenodo deposition (https://zenodo.org/record/8272889), but this is currently embargoed. Please could you make this deposition accessible to me so that I can check your data compliance?

d) Please cite the location of the data clearly in all relevant main and supplementary Figure legends, e.g. “The data underlying this Figure can be found in S1 Data” or “The data underlying this Figure can be found in https://doi.org/10.5281/zenodo.8272889”

e) In the Acknowledgments, you say "Ministry of Environment in Ecuador provided the research permit Nº 0162019ICFLOFAUDNB/MAE required to conduct field work" - please move this information to the appropriate section of the Materials and Methods (presumably "Empirical dataset").

f) I note that your code is currently available in GitHub (https://github.com/f-duchenne/Cheating_in_mutualistic_networks). However, we need this to be in a permanent DOI'd version. Please add it to your Zenodo deposition (or an equivalent) and supply the DOI.

We expect to receive your revised manuscript within two weeks. 

*Published Peer Review History*

*Press*

Sincerely,

Roli Roberts

Roland Roberts, PhD

Senior Editor,

rroberts@plos.org,

PLOS Biology

DATA POLICY:

Regardless of the method selected, please ensure that you provide the individual numerical values that underlie the summary data displayed in the following figure panels as they are essential for readers to assess your analysis and to reproduce it: Figs 2ABCD, 3ABC, 4ABCDE 5, S1AB, S2, S3, S4AB, S5ABC, S6, S7, S8. NOTE: the numerical data provided should include all replicates AND the way in which the plotted mean and errors were derived (it should not present only the mean/average values).

DATA NOT SHOWN?

REVIEWERS' COMMENTS:

Reviewer #1:

Round 2 Review of Duchenne et al. 

I was reviewer 1 in the first round. My line numbers refer to the submitted manuscript (i.e. not the track changes).

In my view, the authors have done an excellent job at revising the manuscript. I remain of the view that it presents interesting and well supported results. While the paper remains undoubtedly highly complex, the authors do a solid job of presenting their results and in this revision the overall narrative of the paper is considerably more clear. I have a few further comments, but they are almost all aimed at improving the clarity further. As such they come with the caveat that much is likely to be personal preference, but hopefully it helpful to the authors to have another view from someone more distant from the work. I've focussed my comments on the figures, as these will be crucial to distilling the complex story being told here. 

42/43 - Not totally clear what is meant by 'in this case' - I think it means 'for innovative cheating', but it could be speed read to mean 'in this simulation'. 

45 - I would suggest getting a couple of key words about your dataset into the abstract, at least mentioning that it is hummingbirds. 

53 - 'described'

58- 'double 'but' in this sentence is not ideal. (maybe switch second to 'although' )

63 'cost to them' (maybe?)

Figure 2 - There is lots of complex information in this figure, and generally I think the authors do a good job at wrestling the multi-dimensional data into some order. One aspect I would question however, is quite which dimension should be in the prime position in the x-axis of the facets. The results text emphasises the proportion of innovative cheating - perhaps this should be on the x?

I also wonder what the value of including cheating frequency =0? In these situations, the average effect on persistence will always be zero (right?) By including it, it creates a somewhat artificial barrier on the left side of each facet. This is not a functional problem, but it adds to the overall cognitive load of the reader. 

d - is this a proportion/fraction of the volume?

176 - > specialists -> specialist

183 - I would consider highlighting this result in the abstract, that the pattern is dependent on a certain level of diversity. 

185 - might be good to comment how long these long-loops are

186 - some agreement issues here. 

Figure 3 - It took me a little while to get that the lower part of b was a zoomed in (I know it says in the caption, but ideally it should be obvious.) I know it must feel unnecessary, but something like writing 'zoom' on the figure could help smooth the journey?

I was not confident in following what 3b is showing - is each dot a particular parameter combination? (changing only the diversity?) What was the grey area? Just the width of the x-axis zoomed in? It might make more sense to highlight the area zoomed? At the moment it looks like it is an area where something special happens. 

If I'm honest, this was also a case there the sheer number of points is not particularly helpful - thinning the data may actually add clarity or perhaps switching to a heat map? As far as I can make out, the story of this plot is quite simple - keeping a 1:1 ratio, and adding some annotations ('small communities impacted more ' / 'large communities impacted more' ) may tell this story in a crisper way. 

3c - same with the other barplot to be honest, it is really hard to see the '0' cases for the generalists. As this is essentially only 4 numbers being presented, could it just be a little table?

206 -> specialist species (and elsewhere)

215 - 'chi 2 = 4,29' Should this be a decimal point '.', or is it describing the degrees of freedom in some way?

Figure 4.

a) I still don't think the web is helpful as presented, but that is ultimately up the authors… 

Adding a colour legend for the country would be helpful. 

b) In the main text, it is described that the there was no statistically significant effect of country, but the plots show a full model including country effects. While not 'wrong' at all, it does make it harder to work out the conclusions we are meant to draw - essentially to shift to thinking that the lines drawn are just to 'guide the eye' and that the confidence intervals are drawn from a maximal model (explaining the uncertainty of direction in some cases). 

I still, it seems like some of the other reviewers, find the part parameterising the theoretical model with the empirical data to be a relatively weak part of the paper. Again, I don't find t wrong as such, just that there isn't a clear conclusion, and it has the effect of adding yet more complication to a challenging paper. But this is ultimately decision for the authors, and I do understand their perspective on this. Their conclusion on L298 (where in parameter space the real system falls) could perhaps be reached more directly than the current analysis. 

310 - is there something to cite in terms of prior expectations?

328 - subtleties ?

340 - can you cite some example previous studies?

Reviewer #2:

The manuscript has improved since the previous submission and the authors have largely addressed the substance of my 1st, 2nd and 3rd major comments. I did find the logic in one of the new sections of the introduction to be hard to follow even after multiple readings but I think this could be addressed with simple changes to the writing and structure, so I have made specific suggestion below. I am happy to leave it to the editor to decide whether the authors are sufficiently clear in any final version.

The authors responded to my 4th comment and to the other reviewers comments re figure 5, by reiterating their justification for this section and it's overall role in the paper. I appreciate the authors clarification but I am still not convinced that they have addressed the substance of our comments; namely that it's not clear what is learnt by this parametrization when the the parameters being varied do not map to measurable features of the system. I understand that the authors want to keep this section of the paper so I have two more suggestion that I think would help clarify things.

Firstly a more minor suggestion; in line 240-241 the authors should state clearly what information from the empirical network is used in the parametrization. They don't need to go into the full detail they have in the methods (which is now much clearer in lines 518-528), but they should state what features of the model are constrained by the empirical data (i.e. network structure and cheating frequency). Right now I think the scope of this analysis is unclear in the main text.

Secondly, in my view, looking at the effect on coexistence whilst varying non-estimable parameters is not sufficient evidence for the claim they are making (Lines 240-241), because there is no reference point against which the parameters values are chosen (even order of magnitude), and there isn't a null model against which the effect on coexistence is being compared (lines 529-533). The authors vary the per-interaction benefit between 1 and 2 and the competition strength at 0.5 and 1 but they could very well have picked any arbitrary interval or scale and this just happened to be a range in which a positive effect is observed. The precise claim that the authors seem to want to make is that the specific network structure and distribution of cheating frequency observed in the empirical network helps promote coexistence compared to some different distribution and a different network structure. To make this claim I think one would have to repeat the analysis on a "randomized" empirical network where cheating frequency and network structure are permuted whilst other parameters are kept constant. If the empirically parameterized model showed higher levels of persistence than this "null model", then I think they can justify their claim that 'that observed cheating patterns increased network persistence, when the cost associated with mutualism was low". Otherwise I don't think this analysis really. proves anything. If both the editor and the other reviewers disagree with my opinion on this, I am happy to be over-ruled as I would prefer not to hold up the entire paper over a single part of the analysis.

Other Comment:

Lines 72-97: This section of the introduction is extremely important and I appreciate the additional details but I found it to be very hard to follow: A few comments might help.

i) Because the entire section is written in the passive voice it introduces a lot of ambiguity and imprecision. For example: 'The consequences of cheating for the number of species being able to coexist (i.e. network or community persistence) are expected to be mediated by two main factors." - expected by who? It's not clear throughout this paragraph what points are established knowledge and what points are the authors reasoning and hypotheses. Simply using the active voice, i.e. "We reasoned that the consequences of cheating for the number of species being able to coexist (i.e. network or community persistence) should be mediated by two main factors" would fix this and improve the flow I would urge the authors to go through the manuscript and try to eliminate this type of ambiguity throughout.

ii) I appreciate that the additional detail was a response to our previous comments but this paragraph is now somewhat meandering and it may be hard for a reader to follow the logic which I think is critically important for understanding this paper. I think it would really help if it were split into 2-3 paragraphs with more explicit signposting of each part of the hypothesis. It is up to the authors how they want to do it but one possible paragraph structure could be: i) Cheating can impact coexistence by two main mechanisms. ii) The effect of cheating on coexistence may depend on whether it is conservative or innovative iii) The effects of innovative cheating may be mediated by network structure and whether the cheater is a generalist or specialist etc.

iii) Punctuation (i.e. commas) throughout this section are inconsistent which made it a little hard read.

Reviewer #3:

The authors have satisfactorily responded to my comments.

---

## [Editor Report · Decision Letter 3]

16 Nov 2023

Dear Dr Duchenne,

Thank you for the submission of your revised Research Article "When cheating turns into a stabilizing mechanism of plant-pollinator communities" for publication in PLOS Biology. On behalf of my colleagues and the Academic Editor, Pedro Jordano, I'm pleased to say that we can in principle accept your manuscript for publication, provided you address any remaining formatting and reporting issues. These will be detailed in an email you should receive within 2-3 business days from our colleagues in the journal operations team; no action is required from you until then. Please note that we will not be able to formally accept your manuscript and schedule it for publication until you have completed any requested changes.

IMPORTANT: The Academic Editor said "It would be great if the authors have good quality photos of the species in their study system, to illustrate the differences between 'conservative' and 'innovative' cheating with examples of the hummingbird/piercers and flower species involved." I would tend to agree, given our broad readership. If you do have some photos that are compatible with our CC BY licence, it might be good to include these in Fig 1. I have asked my colleagues to include such a request with their additional requests mentioned above.

Sincerely, 

Roli Roberts

Senior Editor

PLOS Biology

rroberts@plos.org